# LDL-Cholesterol-Lowering Effects of a Dietary Supplement Containing Onion and Garlic Extract Used in Healthy Volunteers

**DOI:** 10.3390/nu16162811

**Published:** 2024-08-22

**Authors:** Teresa Vezza, Enrique Guillamón, Jorge García-García, Alberto Baños, Nuria Mut-Salud, Jose David García-López, Germán O. Gómez-Fernández, Alba Rodriguez-Nogales, Julio Gálvez, Juristo Fonollá

**Affiliations:** 1Digestive System Service, Virgen de las Nieves University Hospital, 18014 Granada, Spain; tvezza@ibsgranada.es; 2Instituto de Investigación Biosanitaria (ibs.GRANADA), 18012 Granada, Spain; 3DMC Research Center, 18620 Granada, Spain; enrique.guillamon@dmcrc.com (E.G.); abarjona@domca.com (A.B.); nmsalud@dmcrc.com (N.M.-S.); dgarcia@dmcrc.com (J.D.G.-L.); 4Department of Pharmacology, Center for Biomedical Research (CIBM), University of Granada, 18071 Granada, Spain; 5Department of Sciences, Public University of Navarra, 31006 Pamplona, Spain; german.gomez@unavarra.es; 6Centro de Investigación Biomédica en Red-Enfermedades Hepáticas y Digestivas (CIBER-EHD), 28029 Madrid, Spain; 7Department of Nutrition and Food Science, University of Granada, 18071 Granada, Spain; juristo@ugr.es

**Keywords:** hypercholesterolemia, low-density lipoprotein cholesterol, garlic, onion, organosulfur compounds

## Abstract

Hypercholesterolemia plays a pivotal role in the development and progression of cardiovascular diseases, and its prevention seems to be a crucial healthcare strategy to ameliorate these conditions. Subjects with mild hypercholesterolemia are frequently advised against using cholesterol-lowering drugs due to potential side effects, with an emphasis instead on prioritizing dietary adjustments and lifestyle modifications as the primary strategy. In this context, the use of dietary supplements based on medicinal plants may be recommended as a complementary approach to managing elevated cholesterol levels. The aim of this study was to investigate the safety and potential therapeutic effectiveness of a standardized formulation containing extracts from garlic and onions in addressing the health concerns of individuals with slightly elevated cholesterol levels. A controlled, randomized, double-blind, two parallel-group study was conducted over 8 weeks, with clinical visits scheduled at baseline, weeks 2 and 4, as well as at the end of the study. The results revealed significant reductions in both low-density lipoprotein cholesterol and total cholesterol levels among participants who received the extract. Additionally, improvements in blood pressure, as well as in oxidative and inflammatory markers were observed, thus suggesting its potential as a valuable therapeutic intervention for managing mild hypercholesterolemia.

## 1. Introduction

Cardiovascular disease (CVD) is a collective term comprising a group of disorders of the heart and blood vessels. These conditions stand as leading causes of morbidity and mortality worldwide, with important healthcare costs and alarming trends. Indeed, the prevalence of CVD-related deaths has risen from 12.1 million in 1990 to 18.6 million in 2019, with projections suggesting a further increase to 24 million by 2030 [1]. The management of CVD clearly involves the control and/or prevention of the main cardiovascular risk factors, including sedentary lifestyle, smoking, hypertension, diabetes mellitus, overweight/obesity and hypercholesterolemia [2].

Notably, elevated blood concentrations of low-density lipoprotein cholesterol (LDL-C) have been firmly established as causative factors in atherosclerotic disease [3]. Various therapeutic approaches aimed at mitigating hypercholesterolemia, such as statin therapy in conjunction with dietary fat restrictions, have exhibited efficacy in both the treatment and prevention of atherosclerosis [4]. However, due to concerns about the side effects associated with these pharmacological interventions, their use is typically restricted to patients with a significant increase in plasma LDL-C levels or to individuals with multiple CVD risk factors, even if hypercholesterolemia is absent. On the contrary, those subjects with slight hypercholesterolemia are often advised to integrate dietary modifications alongside lifestyle changes. Within dietary strategies, the inclusion of functional foods or food supplements with hypocholesterolemic properties holds potential. This approach aims to maintain cholesterol levels within normal ranges, thereby counteracting the typical upward trend and mitigating the associated deleterious effects.

Natural products play a prominent role as sources of therapeutic agents, particularly those derived from medicinal plants. Among these, garlic (*Allium sativum* L.) and onion (*Allium cepa* L.) are extensively utilized in human nutrition, being renowned for their medicinal properties, which include antioxidant, cardiovascular protective, anticancer, anti-inflammatory, immunomodulatory, anti-diabetic, anti-obesity, and antibacterial effects [5]. All these beneficial properties have been attributed to the presence of various bioactive compounds such as vitamins, minerals, fiber, essential amino acids, phenolic compounds, and organosulfur compounds (OSCs) [5]. OSCs, such as allicin, diallyl mono-, di-, and tri-sulfides, propyl-propane thiosulfinate (PTS), and propyl-propane thiosulfonate (PTSO), among others, are particularly noteworthy. They are believed to play a significant role in the beneficial properties exerted by *Allium* species, particularly in terms of their anti-inflammatory, antioxidant, antidiabetic, and hypocholesterolemic activities [6,7,8]. When considering the latter, the ability of OSCs to positively influence cholesterol production has been reported in different in vitro studies. For instance, in the primary culture of rat hepatocytes, S-allyl cysteine and other organosulfur compounds demonstrated concentration-dependent inhibition of cholesterogenesis through the interaction with the phosphorylation cascade of HMGCoA-reductase [9,10,11,12,13]. In addition, incubation of human THP-1 macrophages with S-allyl cysteine resulted in increased expression of ATP-binding cassette transporter A1 (ABCA1), which plays a pivotal role in cellular cholesterol homeostasis by facilitating the efflux of cholesterol and phospholipids from cells to lipid-poor apolipoproteins, such as apoA-I, in the blood [14]. This process, known as reverse cholesterol transport, is essential for preventing the accumulation of excess cholesterol in cells, particularly in macrophages within arterial walls, which is a key step in the development of atherosclerosis [15].

These observations have been supported by in vivo preclinical studies. For instance, the OSC diallyl tetrasulfide was able to reduce the levels of total cholesterol and LDL-C when administered to Cadmiun-treated rats, an experimental model of dyslipidemia [16]. Similarly, the administration of garlic oil, containing diallyl trisulfide as the main OSCs, to obese rats exhibited beneficial effects by improving various obesity-associated parameters, including cholesterol [17,18]. Moreover, in an experimental model of obesity induced by a high-fat diet in rats, diallyl disulfide, another OSC present in garlic, showed a dose-dependent reduction in plasma levels of triglycerides, total cholesterol, and LDL cholesterol when compared to the untreated control group [19].

Although there are numerous in vivo and in vitro preclinical experiments, knowledge about the effects of these compounds on humans is limited, focusing mainly on garlic. For instance, it has been recently reported that daily consumption of black garlic significantly reduced total cholesterol and LDL cholesterol levels in subjects with hypercholesterolemia, with these effects attributed to its content of OSCs [20]. However, there is a lack of research specifically on the OSCs of onion, and even less is known about the combined effects of garlic and onion. This highlights a significant gap in the scientific literature and underscores the need for more comprehensive studies to explore the potential synergistic effects of these *Allium* derivatives on human health.

Considering these antecedents, in this study, a formulation based on garlic powder and a standardized onion extract with organosulfur compounds derived from propiin was evaluated on healthy subjects with slightly elevated LDL cholesterol levels. The aim was to validate the beneficial effects of these compounds during the early stages of lipid metabolism alteration.

## 2. Materials and Methods

### 2.1. Ethics, Approval, and Consent

This study was performed in accordance with the ethical principles outlined in the Declaration of Helsinki for Medical Research involving human subjects and its amendment, as well as with the Guidelines on Good Clinical Practice standards of CPMP/ICH/135/95 and ISO 14155, along with all relevant Spanish guidelines. The study protocol was reviewed and approved by the Regional Ethical Committee (CEIM/CEI Provincial de Granada, Spain; COA No.C003; 1 October 2020). Additionally, it was registered in the US Library of Medicine (http://www.clinicaltrials.gov (accessed on 22 February 2023)) under the ID NCT04646382.

Volunteers were recruited via email from among the relatives and staff of the Faculties of Pharmacy and Medicine of the University of Granada. All participants provided written informed consent before being enrolled in the study.

### 2.2. Subjects and Study Design

The study enrolled 66 healthy volunteers of both sexes who lived in Granada capital (Spain) and its metropolitan area, aged between 18 and 65 years old, and had LDL cholesterol levels ranging from 100 to 190 mg/dL. LDL cholesterol levels were demonstrated by a recent blood test (performed within the last 3 months). Exclusion criteria included pregnancy, diabetes, cerebrovascular disease, any serious illness, and the use of products or drugs to control cholesterol levels or with antioxidant activity.

A controlled, randomized, double-blind, two parallel-group study was carried out for 8 weeks. Eligible volunteers were assigned a single random number and then randomized equally to the treatment or placebo group, with 33 subjects in each group (33 subjects in the *Allium* group and 33 in the control group). The randomization was performed using a simple random assignment via an online tool that utilized the “Math.random” method from the JavaScript programming language, with the computer clock as the seed. Clinical visits were scheduled at baseline (T0), weeks 2 (T2), and 4 (T4) and at the end of the study (T8) (Figure 1).

### 2.3. Intervention

Participants were randomly assigned to receive either the product or a placebo. The product, Aliocare^®^ (Enzim Orbita Ltda, Olhao, Portugal), contained a concentrated onion extract (86 mg) standardized in organosulfur compounds derived from propiin (10 mg per capsule), garlic powder (14 mg), and microcrystalline cellulose (9892-Capsucel^®^, Laboratorios Guinama, La Pobla de Vallbona, Valencia, Spain) up to 450 mg. The placebo product consisted solely of 450 mg of microcrystalline cellulose. The dosing regimen of one capsule per day was established based on the recommendations of the manufacturer and the findings from a previous study conducted on elderly resident volunteers with infectious respiratory diseases [8].

Both formulations, the active product and the placebo, were delivered in hydroxypropyl methylcellulose capsules (Solchem^®^, Barcelona, Spain). Capsules were dispensed in bottles containing 28 capsules during visits T0 and T4, and participants were instructed to return empty bottles (or bottles with unconsumed capsules) at visits T4 and T8.

Volunteers were instructed to consume one capsule daily during lunch and maintain their regular dietary habits during the intervention. Moreover, the diets of the participants were monitored throughout the study, and they were required to adhere to their usual dietary patterns.

To maintain blinding, neither the researchers involved in the intervention and statistical analysis nor the participants were aware of the treatment assignments.

Considering these circumstances, our research team reached a consensus to establish this dosage. Furthermore, to maintain high-quality standards and ensure the content of active principles, each batch has been rigorously analyzed using advanced chromatography technologies.

### 2.4. Clinical Parameters

Biochemical parameters, including LDL cholesterol, total cholesterol, HDL cholesterol, triglycerides, glucose, alanine aminotransferase (ALT), aspartate aminotransferase (AST), gamma-glutamyl transferase (gamma GT), and creatinine, were analyzed by Reference Laboratory S.L. (Barcelona, Spain) using colorimetric methods. CRP was assessed using the ultrasensitive method by the same laboratory. Other parameters were determined from plasma samples utilizing commercial ELISA kits: IL-1-β, IL-6, IL-10, and malondialdehyde (MDA) from Invitrogen-ThermoFisher Scientific (Waltham, MA, USA), and oxidized LDL from Mercodia (Uppsala, Sweden). Nitric oxide (NO) values were derived from plasma samples by spectrophotometry using nitric oxide assay kits (Invitrogen-ThermoFisher Scientific, USA). All assays were performed in duplicate.

Product acceptance and tolerance were monitored via direct surveys during visits 2, 4, and 8, along with tracking lifestyle habits throughout the study, including physical activity and exercise levels. Dietary intake was assessed through a 7-day food frequency survey at 0, 4, and 8 weeks (Figure 1).

Finally, weight measurements were obtained using the BYH01 balance from WANT Balance Instrument Co. Ltd. (Changzhou, China), while blood pressure readings were taken with an Omron M3 Comfort automatic device (Shimogyō-ku, Japan). These assessments were conducted on the left arm in triplicate, with readings recorded at 10 min intervals.

### 2.5. Statistical Analysis

The sample size calculation and statistical analysis were carried out by the company SEPLIN Soluciones Estadisticas, S.L. (Granada, Spain). The study blind was only opened by the principal investigator upon receipt of the statistical report.

A study was designed to determine the necessary sample size to detect a between-group difference of 15 units in LDL cholesterol units, assuming a standard deviation of 20 units in the study population, with a statistical power of 80% and a significance level of 5%. Based on these considerations, it was determined that 29 patients were required in each group, totaling 58 patients. However, to account for potential losses during the study, 66 volunteers were recruited. The analysis followed the per-protocol approach, with 64 volunteers included and no imputation required due to the completeness of the data.

For descriptive parameters, *t*-tests and Kruskal–Wallis tests were employed with a significance level of 0.05 to assess null differences between the control and intervention groups.

For the analysis of study variables, a two-way repeated measures analysis was performed, with time as the within-subjects factor and treatment as the between-subjects factor. For this purpose, the robust method based on bootstrapping with 10,000 re-samples was applied using the RM function of the MANOVA RM package available in R. Additionally, pairwise comparisons between and within times were performed using *t*-tests with a significance level of 0.05.

## 3. Results

### 3.1. Study Population

The present study aimed to assess the efficacy of a standardized formulation containing a concentrated onion extract and garlic powder in addressing hypercholesterolemia, particularly in subjects at the initial stage of the condition, characterized by plasma LDL cholesterol levels ranging from 100 to 190 mg/dL.

Out of the 2000 volunteers screened initially, 107 expressed interest in participating, and ultimately, 66 subjects were randomized and enrolled in the study. Sixty-four individuals completed the trial successfully from January 2022 to March 2023. Of note, the mean compliance rate to the treatment was above 95% in both groups; indeed, only two participants, one from each group (treatment and placebo), were excluded due to poor compliance.

The enrolled participants ranged in age from 28 to 54, with 32 women and 34 men. A flowchart of the study is depicted in Figure 2.

Table 1 describes the baseline characteristics of the participants in both groups, where no differences between groups were found. Notably, cholesterol levels were slightly higher in the treatment group compared with the control group.

This homogeneity enhances the validity of the study findings by minimizing potential confounding variables and ensuring that observed effects can be attributed to the intervention rather than inherent differences among participants.

Following this, all clinical variables were monitored at different time points: 2, 4, and 8 weeks.

### 3.2. Safety Assessment

The safety profile of the product was assessed through meticulous monitoring of hepatic ALT, AST, and GGT and renal creatinine levels during the trial (Table 2). These biomarkers serve as critical indicators of liver and kidney function, pivotal organs susceptible to toxicity from exogenous substances. Upon analysis of the study data, normal values were observed for the assessed biomarkers, with notable findings even revealing a reduction, especially within the treated group. These results strongly suggest the safety of the treatment. In addition, all participants exhibited excellent tolerance to the extract intake throughout the study duration. There were no reported incidents of adverse events or discomfort associated with dietary supplement consumption, indicating its excellent safety profile. Participants consistently adhered to the prescribed regimen without any difficulties or concerns, underscoring their high level of compliance with the study protocol. Of note, none of the participants were awake during their group assignment, thus indicating that the blinding was appropriately conducted.

### 3.3. Biochemical Outcomes

Throughout the trial, clinical values within the control group remained largely stable, with only minor changes observed in parameters (Table 2). In contrast, participants receiving the treatment displayed notable improvements in some of their clinical parameters within just a few weeks of dietary supplement administration. Firstly, both LDL and total cholesterol levels were significantly reduced compared to the baseline values (T0), as evidenced in Table 2. Notably, LDL data revealed a consistent trend towards reduction, even as early as 2 weeks into treatment, and this reduction became statistically significant at T4 and T8, with decreases of 4% and 6%, respectively. Likewise, total cholesterol exhibited a marked decrease at T4 and T8 (approximately 6%).

Of particular significance is the response observed in volunteers with initial LDL values exceeding 110 mg/dL. These individuals demonstrated a particularly marked reduction in LDL values at both the T4 (5%) and T8 (12%) time points compared to baseline (Table 3).

Furthermore, participants who consumed the product showed a statistically significant decrease of approximately 6% in their total cholesterol levels when comparing measurements from their initial visit to those at their final visit (Table 2).

These notable reductions highlight the rapid effectiveness of the tested formulation in positively modulating cholesterol levels over the duration of the study. In contrast, there was no observable change in HDL cholesterol levels in the group receiving the supplement, while a slight but significant increase was observed in the control group. This clinically insignificant outcome may be attributed to a compensatory mechanism in response to the slight increase in LDL cholesterol values within the same group.

On the other hand, no significant difference was observed in glucose levels, while a significant reduction in plasma triglyceride levels was evident with the treatment (Table 2).

### 3.4. Habits, Dietary and Anthropometric Changes

During the study, volunteers did not make significant changes to their lifestyle or dietary habits.

Despite the absence of significant weight changes observed throughout the trial, notable improvements were observed in systolic and diastolic blood pressure among all participants. Particularly, those subjects receiving the treatment showed the most significant modifications in these parameters at T8 (Table 4), with a reduction of about 5%, indicating the potential cardiovascular benefits associated with the extract.

### 3.5. Oxidative and Inflammatory Parameters

During the intervention period, different variables related to oxidative and inflammatory status were assessed, including oxidized LDL, NO, MDA, IL-1β, IL-10, and IL-6. IL-1β and IL-10 were undetected, as the participants were healthy individuals.

Interestingly, both the treatment and control groups exhibited no significant differences in MDA levels and only minor fluctuations in NO throughout the trial (Table 5). In contrast, participants treated with Aliocare^®^ showed a remarkable reduction in oxidized LDL levels. Notably, significant reductions in oxidized LDL levels, which serve as a marker of oxidative stress and inflammation and are critical in the pathogenesis of cardiovascular diseases, were observed as early as 2 weeks into the treatment, with a further improvement noted at 4 weeks and 8 weeks compared to baseline. Specifically, at T2, oxidized LDL levels decreased approximately 10%, followed by subsequent reductions to 16% at T4 and 32% at T8 when compared to baseline (Table 5). Regarding IL-6, participants who received the supplement consistently displayed a decrease (although not reaching statistical significance) in IL-6 levels compared to those in the control group at the end of the study, although statistical significance was not reached.

## 4. Discussion

Hypercholesterolemia, defined as an elevated blood concentration of LDL cholesterol, is a significant risk factor for atherosclerotic cardiovascular diseases [3,21]. These include peripheral arterial complications, stroke, and myocardial infarction, which continue to be the major causes of premature death, disability, and healthcare expenditure globally. Their prevalence varies by geographic region, ethnicity, and lifestyle factors such as diet and physical activity levels. However, several studies have shown a dramatic increase during the last four decades. Thus, the prevention of hypercholesterolemia is central to any healthcare strategy to ameliorate these conditions.

Nowadays, many cholesterol-lowering drugs demonstrate positive efficacy, but their utility for long-term management is limited by inconvenient dosing schedules, high costs, and concerns about adverse effects [4]. For this reason, additional research approaches are emerging. Among these, the use of alternative and/or complementary treatments, such as dietary supplements based on herbal extracts, is increasing constantly. However, well planned scientific studies focusing on the safety and/or efficacy of these products are limited. For this purpose, the present study aimed to shed light on the safety and potential therapeutic effectiveness of a formulation combining garlic and onion extracts in addressing the health concerns of individuals with slightly elevated cholesterol levels.

Specifically, a controlled, randomized, double-blind, two parallel-group study was carried out for 8 weeks, and clinical visits were scheduled at baseline (T0), weeks 2 (T2) and 4 (T4), and at the end of the study (T8).

During the study, volunteers did not make significant changes to their lifestyle or dietary habits. However, it is important to note that the average BMI values for both the control and treatment groups fall within the overweight range (BMI = 25–29.9). Spain, known for its Mediterranean diet—rich in fruits, vegetables, whole grains, nuts, olive oil, moderate amounts of fish and poultry, and low in red meat and dairy—has been associated with numerous health benefits, including a reduced risk of cardiovascular disease. Nonetheless, recent shifts in lifestyle and dietary habits, especially among younger populations, have led to increased consumption of processed foods, sugary drinks, and fast food, contributing to higher BMI values. Additionally, sedentary behaviors have increased due to urbanization and technological advances. Consequently, the trend towards more Westernized diets and sedentary lifestyles may help explain the observed overweight status among our study participants.

Our findings provide compelling evidence regarding the safety of the treatment, as all safety biomarkers assessed—ALT, AST, gamma GT, and creatinine—remained consistently within normal ranges throughout the duration of the trial in the treated group. Notably, some of these biomarkers even exhibited reductions, suggesting a potential beneficial effect of the treatment on hepatic and overall metabolic function. In addition, participants showed excellent tolerance to the extract intake, with no reported incidents of adverse effects or discomfort. Overall, these results underscore the excellent safety profile of the treatment and alleviate concerns regarding potential adverse effects on liver and kidney function.

Also, it is worth noting that those participants receiving treatment showed reductions in both LDL and total cholesterol levels, thus supporting its potential as a valuable therapeutic intervention for managing dyslipidemia. These beneficial effects were even more evident in those subjects with initial LDL values over 110 mg/dL, thus highlighting the ability of the extract to effectively modulate lipid metabolism and mitigate cardiovascular risk in these circumstances.

These findings are consistent with previous studies. Indeed, both garlic (*Allium sativum* L.) and onion (*Allium cepa* L.) have been largely investigated individually. Thus, the administration of aged black garlic or garlic tablets are able to exert favorable effects on dyslipidemia, as reported by Sobenin et al. [22], who demonstrated the anti-hypercholesterolemic effect of garlic powder, at doses of 300 mg/day, when administered for 12 months. Moreover, this intervention led to marked reductions in LDL cholesterol levels among individuals diagnosed with coronary artery disease. Similarly, subjects with type 2 diabetes mellitus revealed reductions in both total cholesterol and LDL cholesterol levels following a 12-week regimen of consuming 600 mg/day of garlic [23]. In addition, obese patients receiving garlic supplements, at doses of 800 mg/day for 3 months, also showed significant decreases in the same parameters [24]. Finally, a meta-analysis of 39 trials including 2300 participants revealed the ability of garlic to effectively reduce total serum cholesterol and LDL cholesterol in people with high total cholesterol levels (>200 mg/dL), although the plasma triglycerides were not modified significantly [25].

Different randomized controlled trials have also explored the beneficial impact of diets rich in onions on blood lipid levels [26], although the precise mechanisms involved remain elusive. Firstly, onion extracts have been shown to stimulate the excretion of bile acids while simultaneously inhibiting the absorption of cholesterol, leading to a reduction in plasma cholesterol levels [27]. Secondly, onion intake has been observed to activate lecithin–cholesterol acyltransferase, potentially by augmenting insulin activity. This activation may facilitate the conversion of LDL cholesterol into HDL cholesterol [27]. Additionally, this extract has been reported to decrease the concentrations of lipid hydroperoxide and lipoperoxide [28,29].

However, other studies have shown contrasting results, and no significant effects were observed after garlic or onion supplementation on lipid profiles. For example, mild hypercholesterolemic individuals taking 1.4 g/day of garlic powder for 6 months [30], overweight smokers receiving 2.1 g/day of garlic for 3 months [31], and type 2 diabetic patients taking 1.2 g/day of garlic for 4 weeks [32] did not experience significant changes in their lipid profiles. These contradictory findings may be due to variations in study design, duration of supplementation, dosage, and the metabolic status of participants at baseline. Additionally, it is important to consider that the type of garlic or onion supplement utilized (aged extract, powder, oil, or raw extract) may lead to different outcomes due to disparities in bioactive metabolite composition and content. For this reason, the use of standardized products, such as Aliocare^®^, could offer more consistent results.

Of note, all the beneficial properties attributed to these extracts have been linked to the presence of various bioactive compounds such as essential amino acids, vitamins, minerals, phenolic compounds, and organosulfur compounds (OSCs). In this regard, diallyl tetrasulfide, an active OSC, was observed to reduce total cholesterol and LDL cholesterol levels when administered to diabetic or obese rats [16,17,18]. These results suggest that its hypocholesterolemic effect can be associated, at least in part, with decreased hepatic cholesterol synthesis. This mechanism could account for the significant decrease in the levels of the three liver enzymes as well as the reduction in plasma triglyceride levels. Similarly, in an experimental model of obesity induced by a high-fat diet in rats, diallyl disulfide, another OSC present in garlic, showed a dose-dependent reduction in plasma levels of triglycerides, total cholesterol, and LDL cholesterol compared to the untreated control group. Interestingly, evidence suggests that diallyl disulfide can reduce the expression of proprotein convertase subtilisin/kexin type 9 (PCSK9), an important regulator of cholesterol homeostasis, in LPS-treated HepG2 cells, thus contributing to hepatic low-density lipoprotein receptor (LDLR) stabilization and up-regulating the uptake of LDL [33]. In addition, several in vivo studies have highlighted that onion-derived compound propyl- propane thiosulfonate (PTSO) improved lipidic metabolism in mice fed a high-fat diet, reducing total and LDL cholesterol without modifying HDL cholesterol [34]. The mechanisms by which this compound exerts its cholesterol-lowering effects are not fully understood, but they may involve inhibition of cholesterol synthesis in the liver, increased excretion of cholesterol, as well as modulation of cholesterol metabolism-related enzymes.

Regarding blood pressure, our results also highlighted notable improvements in systolic and diastolic blood pressure among participants, particularly those receiving the *Allium* treatment, suggesting additional cardiovascular benefits beyond lipid modulation. Hypertension is a major risk factor for cardiovascular disease, and interventions that effectively lower blood pressure levels can significantly reduce cardiovascular morbidity and mortality.

These results agree with those reported previously. In this regard, a meta-analysis of 12 trials including 553 hypertensive participants concluded that garlic supplements could reduce blood pressure by 5–6 mmHg diastolic and by 8–10 mmHg systolic, similarly to standard blood pressure drugs [35]. This reduction in blood pressure was also linked to a significant decrease in the risk of experiencing cardiovascular events by 16–40%. Similarly, in a previous meta-analysis published in 2015, it was proposed that garlic intake was associated with significant reductions in both systolic and diastolic blood pressure in hypertensive patients [36]. In vivo and in vitro experiments have shown that garlic and OSCs exert their antihypertensive effects by different mechanisms that include the amelioration of arterial remodeling through upregulation of the growth suppressor p27 and the attenuation of ERK 1/2 phosphorylation [37], the reduced synthesis of vasoconstrictor prostanoids [38], the inhibition of the angiotensin-converting enzyme activity [39,40] and the upregulation of the levels and activity of NO, as well as through the generation of hydrogen sulfide [41]. Regarding the latter, it is important to note that oxidative stress and inflammation play key roles in the pathogenesis of atherosclerotic cardiovascular disease, and interventions that target these pathways can be useful in preventing disease progression and improving outcomes. Specifically, an inflammatory state can contribute to the increased oxidation of LDL, which results in blood pressure elevation, impaired vasodilation, and increased vascular resistance. Concurrently, oxidized LDL can also stimulate the production of reactive oxygen species, which can further damage the endothelium, thus promoting vascular dysfunction and subsequently perpetuating the inflammatory cycle [42]. The results from our study suggest that the consumption of Aliocare^®^ treatment reduces levels of oxidized LDL and, although without reaching statistical significance, tends to decrease IL-6 levels, which is consistent with previous studies. In this sense, a randomized, placebo-controlled, parallel-arm, double-blinded trial involving 41 hypercholesterolemic individuals showed that supplementation with aged garlic for 13 weeks inhibited lipid peroxidation and decreased oxidative stress, possibly through the inhibition of myeloperoxidase [43]. Similarly, S-allyl-cysteine (SAC), a compound found in aged garlic extract, has been implicated in preventing endothelial cell damage induced by oxidized LDL through several mechanisms, such as the reduction in intracellular glutathione (GSH) depletion [44]. Furthermore, SAC has been shown to inhibit the activation of nuclear factor-kappa B (NF-κB), a transcription factor involved in inflammation and oxidative stress responses. By inhibiting NF-κB activation, SAC can suppress the expression of pro-inflammatory and pro-oxidant genes, thereby reducing the cellular damage caused by oxidized LDL, TNF-α, or H_2_O_2_ [44]. Moreover, aged garlic extract was found to inhibit inflammation in apolipoprotein E-knockout mice, a well-known model of hyperlipidemia and atherosclerosis. Specifically, the treatment reduced the level of IL-1 receptor-associated kinase 4 and TNF-α while enhancing the activity of adenosine monophosphate-activated protein kinase (AMPK) in the liver [45]. Other dietary supplementations with onion extracts pointed to similar conclusions. In a randomized, double-blind, placebo-controlled, crossover trial, healthy females received onion peel extract for two weeks, underwent a 1-week washout period, and then received the other treatment for an additional two weeks. After two weeks of onion peel extract supplementation, the total cholesterol level, low-density lipoprotein cholesterol level, and atherogenic index significantly decreased, although no changes in the activities of erythrocyte antioxidant enzymes or levels of lipid peroxidation markers were observed following supplementation [46].

Finally, and in terms of safety and tolerability, the excellent compliance observed among participants, coupled with the absence of adverse effects, confirms the favorable safety profile of the product. These findings are crucial for establishing the supplement as a safe and well-tolerated therapeutic option for individuals with hypercholesterolemia or other risk factors.

Moreover, the robust blinding process ensures the integrity and reliability of the study results, minimizing potential biases associated with participant awareness of treatment assignments.

## 5. Conclusions

While numerous studies support the cardiovascular benefits of garlic, there are very few studies validating the properties of onion extracts, with most research focusing on quercetin. Our work highlights the unique aspect of studying organosulfur compounds derived from propiin, which are exclusive to onions and not naturally present in garlic. In this sense, we believe our study provides novel scientific insights. Furthermore, to our knowledge, no studies have combined garlic powder and onion extract in a single capsule, revealing potential synergistic effects and generating interest among consumers and researchers in nutrition and health. This unique combination and its potential benefits add a significant layer of novelty to our work.

The results obtained suggest that this treatment could serve as a promising therapy for managing hypercholesterolemia in healthy subjects with a slight increase in plasma LDL cholesterol levels. Additionally, the observed improvements in blood pressure, oxidative stress, and inflammatory markers support its potential utility. In conclusion, daily consumption of a single capsule containing garlic powder and a concentrated onion extract rich in organosulfur compounds was safe and beneficial for improving cardiovascular health parameters in healthy volunteers with moderate levels of LDL cholesterol.

Therefore, integrating this supplement into a balanced diet may enhance overall cardiovascular health benefits, leveraging the synergistic effects of garlic and onion extracts within the context of a holistic approach to nutrition and wellness.

While these initial findings are promising, it is important to note that there are some limitations. The study was conducted during the COVID-19 pandemic, which presented significant challenges. Many older individuals (below 65 years of age, according to the inclusion criteria) who were initially interested in participating chose not to visit a health center for blood samples due to the associated risks. Consequently, the study predominantly included very young volunteers. This demographic skew resulted in the analyzed values being at the lower end of the desired range. If the volunteers had been recruited under different conditions, the mean values would likely have been higher, and the results would have been more robust.

Future studies will aim to include participants with higher baseline LDL levels, a larger sample size, and a longer intervention period as the next steps in our research plan.

## Figures and Tables

**Figure 1 nutrients-16-02811-f001:**
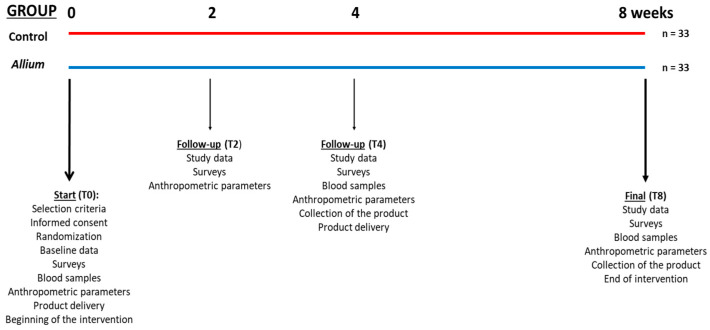
Scheme of the study.

**Figure 2 nutrients-16-02811-f002:**
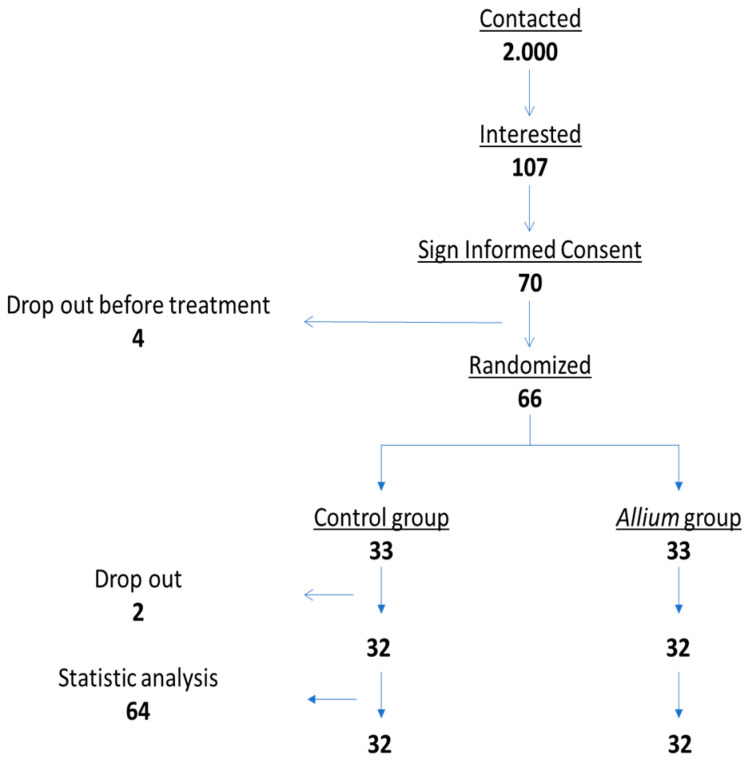
Flowchart of the study.

**Table 1 nutrients-16-02811-t001:** Descriptive values of the study volunteers.

	Control Group (*n* = 33)	Treatment Group (*n* = 33)
Age (years)	39.3 (±11.7)	41.5 (±13.1)
Gender		
Male	18 (54.5%)	16 (48.5%)
Female	15 (45.5%)	17 (51.5%)
Tobacco		
Non-smoker	19 (57.6%)	19 (57.6%)
Smoker	5 (15.2%)	7 (21.2%)
Former smoker	9 (27.3%)	7 (21.2%)
Alcohol		
No drinker	8 (24.2%)	9 (27.3%)
Drinker	25 (75.8%)	24 (72.7%)
Physical activity		
Sedentary	1 (3%)	1 (3%)
Daily homework	10 (30.3%)	13 (39.4%)
Regular activity	14 (42.4%)	15 (45.5%)
Gymnasium	8 (24.2%)	4 (12.1%)
Weight (kg)	76.4 (±18.3)	73.3 (±16.2)
BMI (kg/m^2^)	25.5 (±4.5)	26.4 (±4.4)
Abdominal perimeter (cm)	80.2 (±13.2)	83.2 (±15.0)
Systolic pressure (mmHg)	107.5(±12.8)	110.0 (±13.1)
Diastolic pressure (mmHg)	75.8 (±9.6)	77.1 (±9.5)
LDL cholesterol (mg/dL)	115.4 (±26.2)	122.9 (±29.1)
Total cholesterol (mg/dL)	213.2 (±36.1)	228.9 (±37.5)
HDL cholesterol (mg/dL)	57.8 (±18.2)	59.5 (±15.3)
Triglycerides (mg/dL)	94.5 (±52.7)	106.3 (±59.1)
Glucose (mg/dL)	95.2 (±13.9)	92.5 (±10.9)

Continuous variables are presented as mean ± standard deviation and categorical variables as n (%). No significant differences were found in any of the evaluated descriptive parameters (*p* > 0.05). Detailed descriptions of participants’ tobacco use status, alcohol consumption, and physical activity habits are included in Appendix A.

**Table 2 nutrients-16-02811-t002:** Values of biochemical parameters at 0, 2, 4, and 8 weeks.

	Group	T0	T2	T4	T8	*p* Treatment:Time
LDL cholesterol (mg/dL)	Control	115.4 ± 26.7	115.9 ± 24.3	118.7 ± 30.9	118.5 ± 31.4	0.006
	Treatment	125.0 ± 27.1	123.0 ± 26.5	120.4 ± 27.6 *	117.0 ± 27.0 **^,^†	
Total cholesterol (mg/dL)	Control	213.7 ± 36.6	215.7 ± 34.9	218.1 ± 43.6	217.5 ± 42.2	0.009
	Treatment	231.9 ± 33.8 ^a^	225.9 ± 34.7	220.4 ± 36.3 **	220.2 ± 34.4 **	
HDL cholesterol (mg/dL)	Control	58.3 ± 18.3	60.1 ± 17.3 *	60.8 ± 17.6 *	60.9 ± 18.7 *	0.227
	Treatment	60.3 ± 14.7	60.3 ± 14.6	60.0 ± 14.4	59.7 ± 16.3	
Triglycerides (mg/dL)	Control	88.7 ± 31.2	92.6 ± 31.9	94.6 ± 43.0 †	84.7 ± 32.8	0.071
	Treatment	99.5 ± 54.5	92.1 ± 37.7	86.5 ± 36.1	90.9 ± 38.7	
Glucose (mg/dL)	Control	94.2 ± 12.8	93.3 ± 11.6	93.4 ± 9.7	92.1 ± 11.1	0.962
	Treatment	93.2 ± 10.4	92.7 ± 8.8	92.7 ± 8.3	92.1 ± 10.1	
ALT (U/L)	Control	17.6 ± 8.7	17.5 ± 8.3	18.1 ± 10.1	17.9 ± 10.7	0.220
	Treatment	17.8 ± 8.5	17.2 ± 8.0	15.0 ± 7.5 *	15.7 ± 6.9	
AST (U/L)	Control	21.2 ± 4.7	23.1 ± 14.2	22.9 ± 10.1	23.1 ± 8.9	0.071
	Treatment	25.3 ± 13.5	22.4 ± 6.9	20.1 ± 6.3 *^,^†	23.2 ± 7.6	
Gamma-GT (U/L)	Control	21.3 ± 13.2	21.7 ± 14.4	21.8 ± 13.2	23.4 ± 14.8 ^	0.050
	Treatment	22.4 ± 13.8	21.5 ± 13.8	20.8 ± 12.8 *	20.8 ± 13.2	
Creatinine (mg/dL)	Control	0.96 ± 0.14	0.89 ± 0.22 **	0.92 ± 0.18 *	0.95 ± 0.17 ††	0.505
	Treatment	0.95 ± 0.20	0.92 ± 0.24	0.93 ± 0.24	0.93 ± 0.22	

Values are presented as the mean ± standard deviation. ALT: alanine aminotransferase; AST: aspartate aminotransferase; gamma-GT: gamma-glutamyl transferase. * *p* ≤ 0.05. ** *p* ≤ 0.01 vs. T0; † *p ≤* 0.05. †† *p* ≤ 0.01 vs. T2; ^ *p* ≤ 0.05 vs. T4. ^a^
*p* ≤ 0.05 Control vs. Treatment. Sample: 64 volunteers.

**Table 3 nutrients-16-02811-t003:** LDL cholesterol values at 0, 2, 4, and 8 weeks with volunteers with initial LDL values ≥ 110 mg/dL.

	Group	T0	T2	T4	T8	*p* Treatment:Time
LDL cholesterol (mg/dL)	Control	133.2 ± 21.8	129.6 ± 23.4	137.0 ± 27.5 †	138.2 ± 27.6 ††	0.004
	Treatment	136.5 ± 22.6	133.6 ± 22.8	130.3 ± 24.9 *	126.6 ± 25.5 **^,^†	

Values are presented as the mean ± standard deviation. * *p* ≤ 0.05. ** *p* ≤ 0.01 vs. T0; † *p* ≤ 0.05. †† *p* ≤ 0.01 vs. T2. Sample: 41 volunteers.

**Table 4 nutrients-16-02811-t004:** Values of anthropometric parameters at 0, 4, and 8 weeks.

	Group	T0	T4	T8	*p* Treatment:Time
Weight (kg)	Control	73.3 ± 16.2	72.4 ± 17.4	73.6 ± 16.5	0.672
	Treatment	76.4 ±18.4	76.4 ± 18.2	76.8 ± 18.6 *^,^^	
Systolic pressure (mmHg)	Control	107.5 ± 12.8	104.9 ± 11.5	103.2 ± 12.7 *	0.858
	Treatment	110.0 ±13.1	108.0 ± 13.5	105.1 ± 13.4 ***^,^^	
Diastolic pressure (mmHg)	Control	75.8 ± 9.6	75.3 ± 8.4	73.7 ± 9.1 *	0.518
	Treatment	77.1 ± 9.5	75.0 ± 8.6	73.4 ± 8.0 **	

Values are presented as the mean ± standard deviation. * *p* ≤ 0.05. ** *p* ≤ 0.005. *** *p* ≤ 0.001 vs. T0; ^ *p* ≤ 0.05 vs. T4. Sample: 64 volunteers.

**Table 5 nutrients-16-02811-t005:** Values of oxidative and inflammatory parameters at 0, 2, 4, and 8 weeks.

	Group	T0	T2	T4	T8	*p* Treatment:Time
Oxidized LDL (U/L)	Control	115.4 ± 45.8	118.4 ± 48.6	121.5 ± 47.7	116.1 ± 47.6	≤0.001
	Treatment	143.9 ± 54.9 ^aa^	130.2 ± 56.9 ***	119.4 ± 43.4 **^,^††	98.6 ± 40.8 ***^,^††^,^^^^,^^a^	
MDA (ng/mL)	Control	63.2 ± 49.5	62.1 ± 53.3	69.2 ± 51.3	63.5 ± 44.9	0.247
	Treatment	67.8 ± 60.8	58.4 ± 39.9	59.1 ± 36.9	57.5 ± 35.2	
IL-6 (pg/mL)	Control	4.4 ± 5.7	5.2 ± 7.3	6.9 ± 13.0	6.6 ± 7.9 **	≤0.001
	Treatment	18.4 ± 49.7 ^a^	12.7 ± 31.1	11.3 ± 26.5	8.7 ± 19.6	
Nitric oxide (µM)	Control	33.3 ± 6.1	34.4 ± 4.0	35.2 ± 5.0 *	35.2 ± 5.7	0.646
	Treatment	33.2 ± 4.9	34.4 ± 3.6	34.1 ± 4.3	33.9 ± 4.5	

Values are presented as the mean ± standard deviation. * *p* ≤ 0.05. ** *p* ≤ 0.01. *** *p* ≤ 0.001 vs. T0; †† *p* ≤ 0.01 vs. T2; ^^ *p* ≤ 0.005 vs. T4. ^a^
*p* ≤ 0.05. ^aa^
*p* ≤ 0.005 control vs. treatment. Sample: 64 volunteers.

## Data Availability

Data are contained within the article. The original contributions presented in the study are included in the article/Appendix A, further inquiries can be directed to the corresponding author/s.

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
