# Peer review of "LDL-Cholesterol-Lowering Effects of a Dietary Supplement Containing Onion and Garlic Extract Used in Healthy Volunteers"

_nutrients, 2024, doi:10.3390/nu16162811_

Round 1

Reviewer 1 Report

Comments and Suggestions for Authors

This is an interesting study where authors present the LDL-cholesterol lowering effects of a dietary supplement containing onion and garlic extract in healthy volunteers. Authors capitalize the importance of organosulfur compounds (OSCs) along these lines. It is suggested for authors to provide some additional clarifications:

A)      The identity/concentration of organosulfur compounds (OS) in the tested extracts

B)      Providing that authors have collected blood samples, they could identify OS in the collected samples so as to present a complementary view on the extract active ingredients that can survive the different barriers and remain in the circulation.

C)      Authors should provide with some insights on how the final dosage of the extract was identified.

D)      Authors should indicate the differentiation of the specific work with respect to other relevant works. Indeed, several in vitro studies have demonstrated that garlic compounds can suppress LDL oxidation. Also, human studies have already supported the ability of garlic supplementation to increase the resistance of plasma LDL to copper-induced oxidation. Authors, should indicate the novelty of the current work with respect to the current-state-of-the art.

Comments on the Quality of English Language

Authors should refine the quality of english language

Author Response

This is an interesting study where authors present the LDL-cholesterol lowering effects of a dietary supplement containing onion and garlic extract in healthy volunteers. Authors capitalize the importance of organosulfur compounds (OSCs) along these lines. It is suggested for authors to provide some additional clarifications:

Comment A): The identity/concentration of organosulfur compounds (OS) in the tested extracts

Response A): We thank the reviewer for pointing this out. The organosulfur compounds in question are thiosulfinates and thiosulfonates derived from propiin. We have clarified this in the manuscript.

Lines 137-140: Concentrated onion extract (86 mg), standardized in organosulfur compounds such as thiosulfinate and thiosulfonate derived from propiin (10 mg per capsule).

Comment B):  Providing that authors have collected blood samples; they could identify OS in the collected samples so as to present a complementary view on the extract active ingredients that can survive the different barriers and remain in the circulation.

Response B): Thank you for your valuable comments on our study. Organosulfur compounds (OSCs) are known for their high reactivity and small molecular size, significantly complicating pharmacokinetic studies due to their reactive nature. While our current study did not include the specific analysis you mentioned, our preclinical animal studies employed an analytical platform using GC-MS and UHPLC-Q-TOF-MS to investigate the metabolism of PTSO from onion, one of the main derivatives of propiin included in our dietary supplement. Our findings revealed the presence of dipropyl disulfide (DPDS) in the liver and plasma of the subjects, with the original compound undetected in the blood. However, PTSO and DPDS were excreted in urine 24 hours after ingestion (Garcia-Nicolás et al., 2023).

Furthermore, a human metabolomic study using the same product showed significant changes in the metabolomic profile. This study evaluated the effect of a bioactive dietary supplement on the human plasma metabolome, revealing 26 metabolites affected by the supplement intake. Notably, alterations in phospholipid metabolism were observed, including increases in lysophosphatidylcholines, lysophosphatidylethanolamines, and acylcarnitines, as well as a decrease in the levels of 4 fructosamines. These data suggest that the metabolites exert a biological effect, given the changes observed in plasma analyses. These results align with the antioxidant and antiglycation properties previously associated with Allium extracts, highlighting the potential benefits of the supplement.

Additionally, we have a study under review that further explores this metabolic pathway. Preliminary data indicate that the initial compounds are not detected in plasma or blood but known metabolites such as DPDS from Allium are present.

We hope these clarifications are helpful and remain available for any further questions or comments you may have.

García-Nicolás, M.; Pastor-Belda, M.; Campillo, N.; Rodríguez-Sojo, M.J.; Ruiz-Malagón, A.J.; Hidalgo-García, L.; Abad, P.; de la Torre, J.M.; Guillamón, E.; Baños, A.; et al. Analytical Platform for the Study of Metabolic Pathway of Propyl Propane Thiosulfonate (PTSO) from Allium spp. Foods 2023, 12, 823. https://doi.org/10.3390/foods12040823

Fernández-Ochoa, Á., Borrás-Linares, I., Baños, A., García-López, J. D., Guillamón, E., Nuñez-Lechado, C., Quirantes-Piné, R., & Segura-Carretero, A. (2018). A fingerprinting metabolomic approach reveals deregulation of endogenous metabolites after the intake of a bioactive garlic supplement. Journal of Functional Foods, 49, 137-145. https://doi.org/10.1016/j.jff.2018.10.003

Comment C): Authors should provide with some insights on how the final dosage of the extract was identified.

Response C): We gratefully appreciate this comment. The dosage was established by considering several factors. First, the most suitable doses were determined based on a previous study (cited in the manuscript) conducted on infectious respiratory diseases in elderly resident volunteers. Additionally, lower doses were considered based on toxicological studies. From a practical standpoint, we also determined a consumer tolerance dose, taking into account the characteristic taste and smell of garlic and onion. Despite using a delayed-release capsule, the sensory properties of the concentrated extract influenced the dosage definition.

Considering these circumstances, our research team reached a consensus to establish this dosage. Furthermore, to maintain high-quality standards and ensure the content of active principles, each batch has been rigorously analyzed using advanced chromatography technologies (Cascajosa-Lira et al., 2021).

Cascajosa-Lira, A., Prieto Ortega, A. I., Guzmán-Guillén, R., Cătunescu, G. M., de la Torre, J. M., Guillamón, E., Jos, Á., & Cameán Fernández, A. M. (2021). Simultaneous determination of Allium compounds (Propyl propane thiosulfonate and thiosulfinate) in animal feed using UPLC-MS/MS. Food and Chemical Toxicology, 157, 112619. https://doi.org/10.1016/j.fct.2021.112619

Comment D): Authors should indicate the differentiation of the specific work with respect to other relevant works. Indeed, several in vitro studies have demonstrated that garlic compounds can suppress LDL oxidation. Also, human studies have already supported the ability of garlic supplementation to increase the resistance of plasma LDL to copper-induced oxidation. Authors, should indicate the novelty of the current work with respect to the current-state-of-the art.

Response D): We thank the reviewer for the constructive remark.

While it is true that numerous studies support the cardiovascular benefits of garlic, there are very few studies validating the properties of onion extracts, with most research focusing on quercetin. Our work highlights the unique aspect of studying organosulfur compounds derived from propiin, which are exclusive to onions, as propiin is not naturally present in garlic. In this sense, we believe our study provides novel scientific insights. Furthermore, to our knowledge, no studies have combined garlic powder and onion extract in a single capsule, revealing potential synergistic effects and generating interest among consumers and researchers in nutrition and health. This unique combination and its potential benefits add a significant layer of novelty to our work.

We hope this explanation clarifies the novelty and differentiation of our study and addresses your concerns. The text has been added to the conclusion session.

Thank you for your valuable feedback, and we are open to any further questions or comments you may have.

Reviewer 2 Report

Comments and Suggestions for Authors

The MS entitled LDL-cholesterol lowering effects of a dietary supplement containing onion and garlic extract used in healthy volunteers" is a complex and attentively designed study demonstrating the benefits of Aliocare®regular consumption, 1 cps/daily, without changes in diet and habits. The results are explained involving all cellular and molecular mechanisms of bioactive constituents of both Alium sp. (A. cepa and A. sativum) and compared to those from previous similar studies. 

The following comments are available below:

1. More details are welcome about the participants regarding the following aspects:

- tobacco consumption status: smoker (mild, moderate, or heavy) and former smoker (the period since the participants renounced this habit)

- alcohol consumption - drinker 

- physical activity - Gymnasium

The BMI medium values (in control and treatment groups) belong to the overweight status (BMI = 25 - 29.9); thus, the authors are encouraged to briefly present some important aspects of Spain's dietary habits and lifestyle.

2. The laboratory methods implied in measuring variable parameters and the normal values (different methods lead to different normal ranges).

3. The limitations of the present study should be mentioned.

Author Response

Comment 1:  More details are welcome about the participants regarding the following aspects: tobacco consumption status: smoker (mild, moderate, or heavy) and former smoker (the period since the participants renounced this habit).

Response 1: Thank you for your request for more detailed information about the participants. Below, we provide a breakdown of the participants' tobacco consumption status, alcohol consumption habits, and physical activity levels to give a clearer picture of the study population. We ensured that there was an equal distribution between groups for these characteristics to maintain the integrity and comparability of our study.

  • Smoker:
    • 25% mild (less than 3 cigarettes a day)
    • 50% moderate (less than 10 cigarettes a day)
    • 25% heavy (10 or more cigarettes a day)
  • Former smoker:
    • 95% quit more than 10 years ago
    • 3% quit between 1 to 10 years ago
    • 2% quit less than 1 year ago
  • Alcohol consumption:
    • 95% drink only on weekends and special occasions
    • 5% drink 4 or more days a week
  • Physical activity:
    • 98% exercise 2 or more days a week
    • 2% exercise occasionally or 1 day a week

In the revised manuscript we have provide a table (S1) with all relevant information.

Comment 2:  The BMI medium values (in control and treatment groups) belong to the overweight status (BMI = 25 - 29.9); thus, the authors are encouraged to briefly present some important aspects of Spain's dietary habits and lifestyle.

Response 2:  We acknowledge that the BMI values of our participants fall within the overweight range. To provide context, we would like to share some key aspects of Spain's dietary habits and lifestyle that may influence these findings.

Spain is renowned for its Mediterranean diet, which is typically rich in fruits, vegetables, whole grains, nuts, and olive oil. This diet also includes moderate consumption of fish and poultry, and low consumption of red meat and dairy products. The Mediterranean diet is often associated with numerous health benefits, including reduced risks of cardiovascular diseases and improved metabolic health.

Despite the benefits of the Mediterranean diet, recent changes in lifestyle and dietary patterns have been observed, particularly among younger populations. There is an increasing consumption of processed foods, sugary drinks, and fast food, which contribute to higher caloric intake and, consequently, higher BMI values.

Physical activity levels also play a crucial role in maintaining a healthy weight. In Spain, traditional activities such as walking and outdoor sports are common. However, sedentary behaviors, such as prolonged sitting time and reduced physical activity, are becoming more prevalent due to urbanization and technological advancements.

In summary, while the traditional Mediterranean diet offers numerous health benefits, recent shifts towards more Westernized dietary habits and sedentary lifestyles may contribute to the observed overweight status in our study participants. We hope this provides a clearer understanding of the context within which our study was conducted.

The characteristics of the volunteers recruited for the study met the study criteria: healthy volunteers with a low cardiovascular risk. The diet and habits of the volunteers were monitored throughout the study and were found to be no different from those of the rest of the Spanish population: Mediterranean diet, but with adherence to the same variable. This provides data on obesity and habits that are consistent with those found in our study.

Alejandro Blas, Alberto Garrido, Olcay Unver, Bárbara Willaarts. A comparison of the Mediterranean diet and current food consumption patterns in Spain from a nutritional and water perspective. Science of the Total Environment, 2019. https://doi.org/10.1016/j.scitotenv.2019.02.111

https://www.ine.es/ss/Satellite?L=es_ES&c=INESeccion_C&cid=1259926457058&p=1254735110672&pagename=ProductosYServicios/PYSLayout

Thank you for your insightful comment. We have added the key aspects of Spain's dietary habits and lifestyle in the section “Habits, Dietary and Anthropometric Changes”.

Comment 3. The laboratory methods implied in measuring variable parameters and the normal values (different methods lead to different normal ranges).

Response 3. To avoid methodological bias in the analysis of the samples, the study was designed to measure the parameters using the same method for all volunteers. The same analysis kit was used, and all measurements were performed by the same technician. Additionally, it was ensured that all four samples from the same volunteer were measured at the same time.

Comment 4. The limitations of the present study should be mentioned.

Response 4. The study was conducted during the COVID-19 pandemic, which presented significant challenges. Many older individuals (below 65 years of age, according to the inclusion criteria) who were initially interested in participating chose not to visit a health center for blood samples due to the associated risks. Consequently, the study predominantly included very young volunteers. This demographic skew resulted in the analyzed values being at the lower end of the desired range. If the volunteers had been recruited under different conditions, the mean values would likely have been higher, and the results would have been more robust, as discussed in the article. Future studies will aim to include participants with higher baseline LDL levels, a larger sample size, and a longer intervention period as the next steps in our research plan.

The text in the revised manuscript has been modified to include these limitations in the conclusion section.